# Establishment of Mucoepidermoid Carcinoma Cell Lines from Surgical and Recurrence Biopsy Specimens

**DOI:** 10.3390/ijms24021722

**Published:** 2023-01-15

**Authors:** Shunpei Yamanaka, Susumu Suzuki, Hideaki Ito, Karnan Sivasundaram, Ichiro Hanamura, Ikuko Okubo, Kazuhiro Yoshikawa, Shoya Ono, Taishi Takahara, Akira Satou, Toyonori Tsuzuki, Ryuzo Ueda, Tetsuya Ogawa, Yasushi Fujimoto

**Affiliations:** 1Department of Otorhinolaryngology, Aichi Medical University School of Medicine, Nagakute 480-1195, Japan; 2Research Creation Support Center, Aichi Medical University, Nagakute 480-1195, Japan; 3Department of Tumor Immunology, Aichi Medical University School of Medicine, Nagakute 480-1195, Japan; 4Department of Pathology, Aichi Medical University School of Medicine, Nagakute 480-1195, Japan; 5Department of Biochemistry, Aichi Medical University, Nagakute 480-1195, Japan; 6Division of Hematology, Department of Internal Medicine, Aichi Medical University, Nagakute 480-1195, Japan; 7Department of Maxillofacial Surgery, Aichi Gakuin University School of Dentistry, Nagoya 464-8651, Japan; 8Department of Surgical Pathology, Aichi Medical University Hospital, Nagakute 480-1195, Japan

**Keywords:** head and neck mucoepidermoid carcinoma, karyotype, chimeric gene, cisplatin, 5-fluorouracil, cetuximab, EGFR, Multi-color FISH

## Abstract

Patients with advanced/recurrent mucoepidermoid carcinoma (MEC) have a poor prognosis. This study aimed to establish and characterize human mucoepidermoid carcinoma cell lines from the initial surgical specimen and biopsy specimen upon recurrence from the same patient to provide a resource for MEC research. MEC specimens from the initial surgical procedure and biopsy upon recurrence were used to establish cell lines. The established cell lines were cytogenetically characterized using multi-color fluorescence in situ hybridization and detection, and the sequence of the *CRTC1-MAML2* chimeric gene was determined. Furthermore, the susceptibility of head and neck mucoepidermoid carcinoma to standard treatment drugs such as cisplatin, 5-fluorouracil, and cetuximab was investigated. We successfully established unique MEC cell lines, AMU-MEC1, from an initial surgical specimen and AMU-MEC1-R1 and AMU-MEC1-R2 from the recurrent biopsy specimen in the same patient. These cell lines exhibited epithelial morphology and developed in vitro-like cobblestones. They shared eight chromosomal abnormalities, including der(19)ins(19;11)(p13;?), which resulted in a chimeric *CRTC1-MAML2* gene, indicating the same origin of the cell lines. The susceptibility of all cell lines to cisplatin and 5-fluorouracil was low. Interestingly, EGFR dependency for cell growth decreased in AMU-MEC-R1 and AMU-MEC-R2 but was retained in AMU-MEC1. These cytogenetic and biochemical findings suggest that the established cell lines can be used to investigate the disease progression mechanisms and develop novel therapeutics for MEC.

## 1. Introduction

Mucoepidermoid carcinoma (MEC) accounts for 5–15% of all salivary gland malignancies but less than 1% of all head and neck cancers [1,2]. MEC is a heterogeneous tumor composed of various proportions of mucin-secreting, epidermoid, and intermediate cells. The composing cell proportion is associated with the histological grades. High-grade tumors have a high proportion of epidermoid cells and a relatively low proportion of mucin-secreting cells, whereas low-grade tumors have mucin-secreting cells in more than half of the tumor mass [3,4,5]. Although MEC disease progression is dormant, the 5-year survival rate for high-grade and some intermediate-grade tumors is less than 50%, making them far less likely to be cured than low-grade cases [6,7]. In >50% of MEC, a chromosomal translocation t(11;19)(q14-21;p12-13) is detected that joins exon 1 of the cAMP response element binding (CREB) protein binding domain of *CRTC1* (CREB regulated transcription coactivator 1 [also called *MECT1*, *TORC1*, and *WAMP1*]) gene at 19p13 or CRTC3 gene at 15q26 in-frame to exons 2–5 of the Notch coactivator mastermind like gene 2 (*MAML2*) gene at 11q21 resulting in the expression of a novel *CRTC1/3-MAML2* fusion gene [8,9,10,11]. *CRTC1/3-MAML2* fusion genes are more prevalent in low- and intermediate-grade tumors than in high-grade tumors [9,10,11], and the fusion gene-positive cases have a higher survival rate [9,10,12,13]. The CREB-binding domain of CRTC1 (42 aa) is fused to the transcriptional activation domain of MAML2 (983 aa) to form the chimeric protein CRTC1/3-MAML2, which activates CREB target gene transcription and promotes MEC growth and survival [8,14]. CRTC1-MAML2 co-activates CREB to upregulate the epidermal growth factor receptor (EGFR) ligand, amphiregulin (AREG) [15]. Aberrant AREG-EGFR signaling is critical for the oncogenic function of CRTC1-MAML2. The CRTC1-MAML2 fusion is an oncogenic driver, according to a recent in vivo investigation employing a conditional transgenic mouse model [16].

Only surgery is the standard treatment for MEC, though radiation is sometimes used as postoperative adjuvant therapy for advanced cases. Chemotherapy and radiation therapy can be combined, although the overall survival is not improved compared to radiation alone [17,18,19,20]. Although novel therapeutics for MEC targeting CRTC1/3-MAML2 are being explored, there is currently no viable alternative to surgery [21]. Especially, novel therapeutics for postoperative recurrent cases are needed. Cell lines are invaluable research tools for understanding the molecular biology of MEC cells and developing novel therapeutics. Although several MEC-derived cell lines have been established to date, very few cell lines are publicly accessible. We aimed to establish cell lines from both surgical and recurrent biopsy specimens of the same MEC patient and describe their cytogenetic and biological characteristics.

## 2. Results

### 2.1. Morphological Features of the Established MEC Cell Lines

Phase contrast microscopy images of the MEC cell lines are depicted (Figure 1). All three cell lines, AMU-MEC1, AMU-MEC1-R1, and AMU-MEC1-R2, have similar epithelial cell-like morphology and display cobblestone-like shapes in tissue culture plates. 

### 2.2. Comparison of Karyotypes among the Three Cell Lines

The karyotypes of eight randomly selected cells from each cell line are listed in Appendix A. Nearly identical karyotypes were found in all eight cells of each cell line, indicating their clonality. Representative multi-color FISH images of cells 3, 4, and 5 from AMU-MEC1, AMU-MEC1-R1, and AMU-MEC1-R2 are depicted (Figure 2). AMU-MEC1 and AMU-MEC-R2 were both hyper-diploid, whereas AMU-MEC-R1 was near-diploid (Figure 2A–C). All three cell lines had der(19)ins(19;11)(p13;?) (Figure 2D–F) resulting in *CRTC1-MAML2* chimeric gene (detailed in the next section). Each of the three cell lines, AMU-MEC1, AMU-MEC-R1, and AMU-MEC-R2, shared seven distinct chromosomal abnormalities, including der(19)ins(19;11)(p13;?) (Appendix A), suggesting that all three cell lines originated from the same cell. Both AMU-MEC1 and AMU-MEC1-R2 shared the translocation, [der(9)t(9;13)(p?13;q21)] (Appendix A). The unique abnormalities in AMU-MEC1, AMU-MEC1-R1, and AMU-MEC1-R2 are shown respectively (Appendix A). 

### 2.3. der(19)ins(19;11)(p13;?) Resulted in CRTC1-MAML2 Chimeric Gene Generation

All MEC cell lines displayed RT-PCR amplification of a 194bp gene product, which was consistent with the predicted length of the forward and reverse primers (Figure 3A,B). The same length product was detected in NCI-H292 cell, which had a t(11;19) mutation resulting in a *CRTC1-MAML1* chimeric gene [8], but not in HSC-3, an oral squamous cell carcinoma cell line used as a negative control. In addition, the RT-PCR amplicon sequence analysis verified that the sequences on both sides of the fusion site in the *CRTC1-MAML1* chimeric gene corresponded to the previously reported sequence (GenBank: AY040324.1) [8] (Figure 3C). Thus, all established cell lines in this investigation with der(19)ins(19;11)(p13;?) generated the *CRTC1-MAML2* chimeric gene. 

### 2.4. Growth Activity Analysis

BrdU incorporation rates in the MEC and OSCC cell lines were compared every four days following passaging. MEC cell lines had rates between 1–5%, significantly lower than OSCC cell lines (20–50%). Thus, MEC cell lines exhibited extremely limited growth (Figure 4). 

### 2.5. Comparison of Susceptibility of MEC and OSCC Cell Lines to CDDP and 5-FU

OSCC cell lines showed a drug concentration-dependent reduction in viability. IC50 of CDDP in HSC-2, HSC-3, and HSC-4 was approximately 6.0 µM, 2.0 µM, and 4.8 µM, respectively. The IC50 of 5-FU in HSC-2, HSC-3, and HSC-4 was approximately 2.1 µM, <0.3 µM, and <0.3 µM, respectively. However, MEC cell lines maintained 60–100% viability even at high CDDP and 5-FU concentrations (Figure 5). Thus, the MEC cell lines were highly resistant to the standard OSCC treatments, CDDP and 5-FU.

Cetuximab blocked EGF-EGFR signaling but not proliferation in AMU-MEC1-R1 and AMU-MEC1-R2 cells in all MEC cell lines treated with exogenous EGF; cetuximab inhibited EGFR phosphorylation. Cetuximab also suppressed the EGFR downstream protein, STAT1 phosphorylation (Figure 6A). These findings indicate that cetuximab blocked EGF-EGFR signaling in all MEC cell lines. In addition, cetuximab significantly inhibited BrdU incorporation in AMU-MEC1 but not in AMU-MEC1-R1 and AMU-MEC1-R2 (Figure 6B,C). The EGF proliferation dependency of MEC cells might have been lost when the disease relapsed in the patient. 

### 2.6. PD-L1 Expression in the MEC Cell Lines

PD-L1 expression was not shown in all MEC cell lines. IFN-γ induced PD-L1 expression in AMU-MEC1 and AMU-MEC1-R1 but not in AMU-MEC1-R2 (Figure 7).

## 3. Discussion

We established the AMU-MEC1 cell line from the surgical specimen of a MEC patient and the AMU-MEC1-R1 and AMU-MEC1-R2 cell lines from biopsy specimens at the time of recurrence of the same patient. The MEC-specific *CRTC1-MAML2* chimera is typically caused by the t(11;19)(q14-21;p12-13) translocation [8,14]. Although all the three cell lines had *CRTC1-MAML2,* their chromosomal structural abnormality was der(19)ins(19;11)(p13;?) instead of t(11;19)(q14-21;p12-13). This is the first report of MEC with der(19)ins(19;11)-generated *CRTC1-MAML1*. However, this investigation did not compare the biological and clinical aspects of translocation and insertion cases. These investigations need to be approached in the future. Low- and intermediate-grade patients have more *CRTC1-MAML2* chimeric gene-positive cases than high-grade cases [9,10,11]. However, the established cell lines in this study were derived from a high-grade case. MEC cell lines were previously established from low- (HCM-MEC010) [22], intermediate- (UM-HMC-1, UM-HMC-2, and UM-HMC-3A, B) [23], or high-grade cases (UT-MUC-1 and HTB-41) [24,25]. H292 and H3118 are also MEC cell lines [25,26]; however, the histological grades of the original tumors have not been reported. Thus, the number of cell lines established from high-grade cases with low CRTC1-MAML2 incidence is relatively high, indicating that high-grade cases are highly malignant even if they are *CRTC1-MAML2* positive. Comparisons of these different histological-grade cell lines could be beneficial in the future. 

Many abnormal chromosomal structures were observed, but only seven of them were common among all three cell lines (Figure 4). This finding indicates that the three cell lines have the same origin. Although the prognosis for MEC patients with a *CRTC1-MAML2* chimeric gene is favorable, there is currently no alternative treatment for situations where the disease has worsened or recurred [21]. Therefore, establishing MEC cell lines derived from the initial surgery and recurrent biopsy specimens upon recurrent is critical for understanding disease progression and developing novel therapeutics. The conventional OSCC treatments CDDP and 5-FU were ineffective against all three MEC cell lines. This result is consistent with poor chemotherapeutic outcomes for MEC patients [27]. Additionally, these cell lines were resistant to docetaxel and gemcitabine. Therefore, one possible explanation for the MEC cell resistance to cytotoxic chemotherapies could be that all three MEC cell lines had significantly lower growth activity than OSCC cell lines. 

MEC tumorigenesis is related to abnormal EGFR signaling [15,16,28]. Cetuximab inhibits MEC cell proliferation by blocking EGFR. However, MEC cells are not completely eliminated; instead, some cells survive [15,29]. In our study, growth inhibition by cetuximab was lower in AMU-MEC1-R1 and R2 than in AMU-MEC1. However, cetuximab inhibited EGF-induced EGFR phosphorylation in all cell lines (Figure 6A). This suggests that the MEC cells lost EGFR signaling growth dependence upon the recurrence. Exome analysis revealed a significant rise in the number of gene mutations in the recurring biopsy specimen compared to the initial surgical specimen (see Appendix A). AMU-MEC1-R1 and R2 cells had eight distinct chromosomal structural abnormalities (Appendix A). The accumulation of gene alterations produced by clonal evolution could result in a cetuximab-resistant MEC clone. Our previous study showed that EGFR signaling in OSCCs reduces effector T-cell infiltration and increases Treg, generating an immunosuppressive tumor microenvironment similar to that observed in non-small cell lung cancers [30,31]. Future investigations should focus on the influence of EGFR signaling blockade on tumor immunity and survival using the MEC cell lines. Our established cell lines might be valuable for unveiling additional signaling pathways critical for MEC growth and survival and exploring potential blocking strategies. 

Immunotherapy using immune checkpoint inhibitors (ICIs) has been extensively used for various tumors. However, few clinical trials and basic research have focused on MEC immunotherapy, and its effects are unknown [21]. Recently, an advanced high-grade MEC patient was completely cured by first-line pembrolizumab treatment [32]. This provides a possibility of a new therapeutic option for immunotherapy. However, despite a high cumulative positive score, the patient in this study did not respond to pembrolizumab (CPS: 35–45). Although the association of tumor mutation burden (TMB) with treatment effectiveness for ICIs in MEC is unknown, the KEYNOTE-158 trials report that a subpopulation of 791 patients, including 82 salivary tumors with high-TMB (TMB > 10mut/Mb), showed a higher overall response rate than low-TMB cases [33]. TMB of the patient upon recurrence was 4.6 mut/Mb (see Appendix A), which is relatively low. The upregulation of PD-L1 in response to IFN-γ stimulation varied between the established cell lines. The upregulation was observed in AMU-MEC1 and AMU-MEC1-R1 but not in AMU-MEC1-R2. This suggests that the immune response became more diverse during recurrence. These findings could be the reason for the unfavorable treatment outcome. Therefore, it is desirable to investigate biomarkers for classifying MEC subpopulations that are effective for ICIs. 

The three MEC cell lines established in this study are distinct in three ways, as described below. 1) The cell lines were derived from both surgical and biopsy specimens upon recurrence in the same MEC patient, 2) all cell lines generate the *CRTC1-MAML2* chimeric gene as a result of der(19)ins(19;11)(p13;?) instead of t(11;19)(q14-21;p12-13), and 3) the response to EGF or IFN-γ was heterogeneous among these three cell lines. MEC is a rare cancer with a favorable prognosis. However, there are few treatment options for advanced/recurrent cases, necessitating the development of novel targeted therapies. These cells could be used to study the mechanisms of MEC tumorigenesis and progression and to develop novel therapeutics. 

## 4. Materials and Methods

### 4.1. Patient

The patient is a 69-year-old male with a history of hypertension, hyperlipidemia, pneumonia, and herpes zoster. His father, sister, and aunt had a history of malignant lymphoma. Two months ago, he presented to the otorhinolaryngology clinic with a chief complaint of pain on swallowing and bilateral cervical lymphadenopathy. A laryngeal fiberoptic scan revealed a neoplastic lesion in close proximity to the epiglottis (Appendix A). After biopsy and imaging evaluation, MEC (T4aN2cM0) was diagnosed. Approximately one and a half months after the initial otorhinolaryngological examination, the patient underwent total tongue and laryngeal excision, bilateral neck dissection, and pectoralis major skin graft surgery. Postoperative pathology revealed high-grade MEC (Appendix A) with negative margins and no lymph node involvement. The postoperative course was good, but the recurrence of MEC around the tracheal foramen was diagnosed two months later through biopsy (Appendix A). Radiation (40 Gy) and pembrolizumab (200 mg/body) were administered for one course. Then, two courses of pembrolizumab were administered as monotherapy. Radiation and pembrolizumab monotherapy could not prevent the progression of the recurrent lesions.

### 4.2. Tumor Specimens and Establishment of MEC Cell Lines

MEC cell line, AMU-MEC1 was derived from surgically-resected salivary MEC tissue on 21 May 2019, while AMU-MEC1-R1 and AMU-MEC1-R2 were derived from recurrent carcinoma biopsy specimens on 15 October 2019 (Appendix A). The following procedure was followed to establish the cell lines. First, tissues were minced into small pieces and placed on a culture dish containing Dulbecco’s Modified Eagle Medium (DMEM) (Fuji Film, Tokyo, Japan) supplemented with 10% fetal bovine serum (FBS HyClone Laboratories, Inc., South Logan, UT, USA) and 1% penicillin-streptomycin (Gibco, Grand Island, NY, USA). The carcinoma cells formed a monolayer on the plate during culture. After two months, the cells were detached using trypsin-EDTA (Gibco). Fibroblast and non-target cells were removed via magnetic separation with anti-epithelial cell adhesion molecule magnetic beads using autoMACS (Miltenyi Biotec, Bergisch Gladbach, Germany). The isolated cells were cultured continuously for six months with weekly media changes. Cells passaged ten or more times were used for the experiments. This study was conducted in accordance with the Declaration of Helsinki and approved by the Ethical Committee of Aichi Medical University (approval numbers: 2020-H033 and 2020-H073).

### 4.3. Measurement of Drug Susceptibility Using Water-Soluble Tetrazolium-1 (WST-1) Assay

After MEC and OSCC cells suspended in DMEM (10% FBS) were seeded in 96-well plates at 10^5^ and 2 × 10^4^ cells per well, respectively, serial concentrations of cisplatin (CDDP) (Nippon Kayaku Co., Ltd., Tokyo, Japan) or 5-fluorouracil (5-FU) (Kyowa Kirin Co., Ltd., Tokyo, Japan) were added in triplicate and cultured for five days. The supernatant in each well was completely aspirated, and the cells were incubated with 100 μL of WST-1 solution (Takara Bio, Otsu, Japan) diluted with DMEM (10% FBS) at 37 °C for 2 h. Absorbance was measured at 450 and 620 nm using a microplate reader (BIO-RAD, Hercules, CA, USA). Cell viability was calculated using the following formula: % viability = 100 × (E − S)/(M − S), where E, M, and S denote the absorbances of the experimental well, well without drugs (cells incubated with medium alone), and well with medium alone (without cells). 

### 4.4. Analysis of Bromodeoxyuridine (BrdU) Incorporation and Detection of PD-L1 by Flow Cytometry

Two × 10^5^ MEC and OSCC cells were seeded in 24-well plates and cultured for four days in DMEM (10% FBS). After seeding (days 1–4), the cells were treated with 10 μM BrdU (Fuji Film) and incubated at 37 °C for 1 h. After fixing cells with 70% ethanol at −20 °C for 30 min, double-stranded DNA was denatured with 2N HCl at room temperature for 30 min. After washing three times with buffer (PBS containing 0.2% HSA and 2mM EDTA), the cells were incubated with FITC-conjugated anti-BrdU monoclonal antibody (BD Bioscience, Franklin Lakes, NJ, USA) at 4 °C for 30 min. The cells were washed and resuspended in 1 μg/mL propidium iodide-containing washing buffer. For detection of PD-L1, 2 × 10^5^ MEC and OSCC cells were seeded to 24-well plates, cultured for two days in DMEM (10% FBS), and treated with 25 ng/mL of interferon (IFN)-γ for two days. The cells were collected and incubated with PE-conjugated anti-PD-L1 monoclonal antibody (MBL, Tokyo, Japan) at 4 °C for 20 min. Then, flow cytometry was performed using BD LSRFortessa flow cytometer (BD Biosciences), and the data were analyzed using the FlowJo software v10.8.1 (Tree Star, Inc., Ashland, OH, USA). 

### 4.5. Western Blot Analysis of EGFR Expression and Phosphorylation

The cell lysates in protein extraction buffer (2% Triton X in 10 mM Tris-HCl, 150 mM NaCl, 2 mM ethylenediaminetetraacetic acid (EDTA), and 2 mM 2-mercaptoethanol (pH 7.4)) were resolved on SDS-PAGE gels (10^5^ cells/lane) and transferred to polyvinylidene fluoride membranes using an iBlot 2 Dry Blotting System (Invitrogen, Carlsbad, CA, USA). After blocking in tris-buffered saline (TBS) with 1% skim milk for 1 h at room temperature, the membranes were incubated with primary antibodies: EGFR (1:1000 dilution, MBL), p-EGFR (Y1068, 1:1000 dilution, Cell Signaling Technology, Danvers, MA, USA), p-STAT1 (Y701, 1:1000 dilution, Cell Signaling Technology), and β-actin (1:1000 dilution, MBL). β-actin was used as the internal control. After four TBS washes, the membranes were incubated with peroxidase polymer anti-mouse IgG or anti-rabbit IgG (Vector Laboratories, Burlingame, CA, USA) at a 1:100 dilution for 30 min at room temperature. The chemiluminescence signal was detected using ECL Prime Western blotting Detection Reagent (GE Healthcare Systems, Chicago, IL, USA) and captured using an Amersham Imager 600 (GE Healthcare Systems). 

### 4.6. Karyotype Analysis Using Multi-Color FISH

Chromocenter Co., Ltd. (Yonago, Japan) was approached to conduct the staining and analysis. Cells were sequentially treated with 0.2 µg/mL colcemid for 3 h and 75 mM KCl for 10 min, then fixed using Carnoy’s solution. The fixed cells were denatured with 2 × SSC at 72 °C for 30 min and hybridized with the 24XCyte Human multicolor FISH probe (Metasystem, Reggio Emilia, Italy) at 37 °C for 42 h. Images were captured using a Zeiss Axio Imager. Z2 microscope (Zeiss, Jana, Gernamy). 

### 4.7. Detection and Nucleotide Sequencing of CRTC1-MAML2 Chimeric Gene

MEC cells were plated in 100 mm^3^ dishes, and 5 × 10^6^ cells were collected per 1.5 mL tube. RNA was extracted using the Nucleo Spin RNA Kit (MACHEREY-NAGAL, Dueren, Germany), and Reverse transcription (RT)-PCR was performed using the Prime Script One-Step RT-PCR Kit ver.2 (TAKARA, Kusatsu, Japan) according to the manufacturer’s instructions. A 194-base-pair amplicon was generated to detect the CRCT1-MAML2 chimera using the primer sequences 5′-ATG GCG ACT TCG AAC AAT CCG CGG AA-3′(forward) and 5′-CCA TTG GGT CGC TTG CTG TTG GCA GGA G-3′(reverse). The amplified RT-PCR products of the *CRCT1-MAML2* chimera were detected using 2% agarose gel electrophoresis. PCR products were sequenced using an ABI Prism 377 Genetic Analyzer (Applied Biosystems, Foster City, CA, USA).

### 4.8. Statistical Analysis

Differences between the two groups were examined with the Student *t*-test. *p* < 0.05 was considered statistically significant.

## Figures and Tables

**Figure 1 ijms-24-01722-f001:**
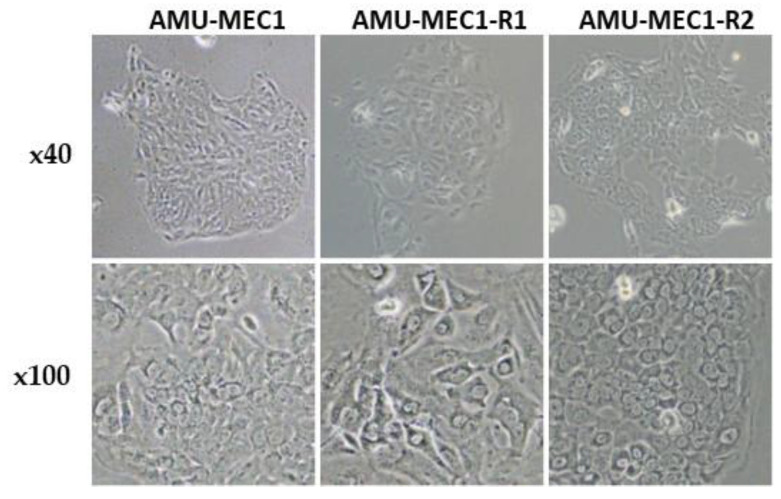
The morphology of the AMU-MEC1, AMU-MEC1-R1, and AMU-MEC1-R2 cell lines. Microscopic images of the MEC cell lines are shown at magnifications of ×40 (upper panels) and ×100 (lower panels), respectively.

**Figure 2 ijms-24-01722-f002:**
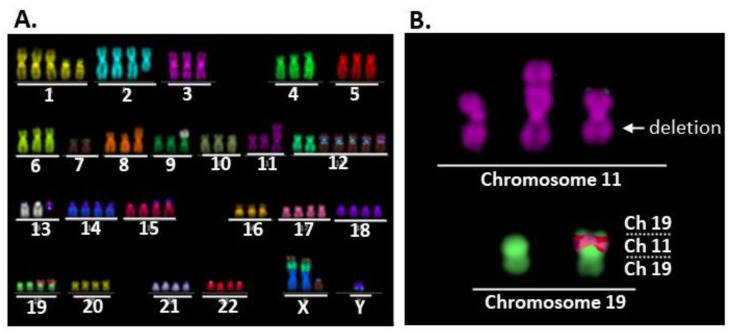
Representative images of multi-color FISH and structural abnormalities of chromosomes 11 and 19 in the established MEC cell lines, AMU-MEC1, AMU-MEC1-R1, and AMU-MEC1-R2. Images of (**A**,**B**) AMU-MEC1, (**C**,**D**) AMU-MEC1-R1, and (**E**,**F**) AMU-MEC1-R2. AMU-MEC1 and AMU-MEC-R2 are hyper-diploid, while AMU-MEC-R1 is near-diploid (**A**,**C**,**E**). All three cell lines have der(19)ins(19;11)(p13;?), resulting in the *CRTC1-MAML2* chimeric gene (**B**,**D**,**F**). White arrows indicate internal deletion loci in the chromosomal arm 11q (**B**,**D**,**F**). The karyotype of each cell line is detailed in Appendix A. Ch, chromosome.

**Figure 3 ijms-24-01722-f003:**
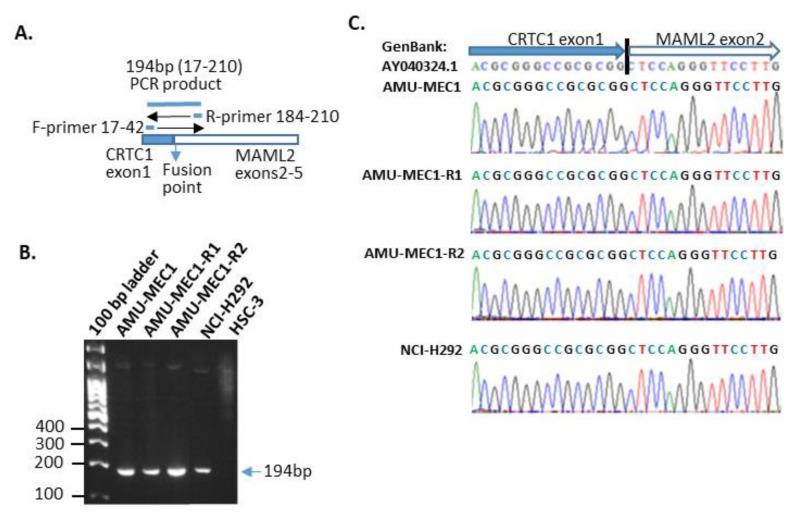
Detection and sequencing of RT-PCR amplified *CRTC1-MAML2* chimeric gene. (**A**) The scheme of *CRTC1-MAML2* chimeric gene amplification. The primer binding site was designed to span the mRNA fusion point and generate a 194 bp amplicon. (**B**) Chimeric gene detection in the RT-PCR products from each cell. (**C**) Sequence analysis of RT-PCR products from each cell.

**Figure 4 ijms-24-01722-f004:**
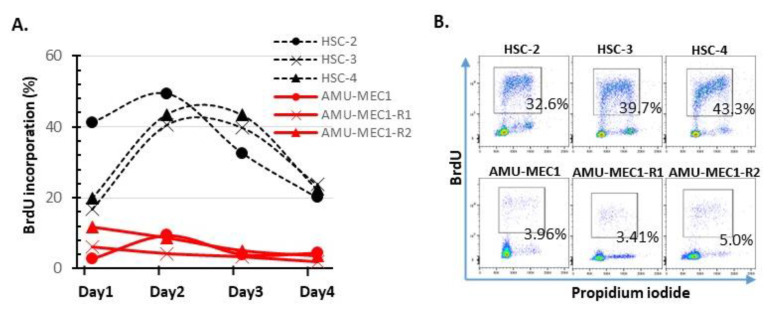
Comparison of BrdU incorporation between MEC and OSCC cell lines. On days 1–4, following cell replating, BrdU incorporation was analyzed using flow cytometry. (**A**) Change in BrdU incorporation (%) in each cell. The red and black dotted lines indicate MEC and OSCC cell lines, respectively. (**B**) Representative cytograms analyzed on day 3. The upper and lower column indicate OSCC and MEC cell lines, respectively.

**Figure 5 ijms-24-01722-f005:**
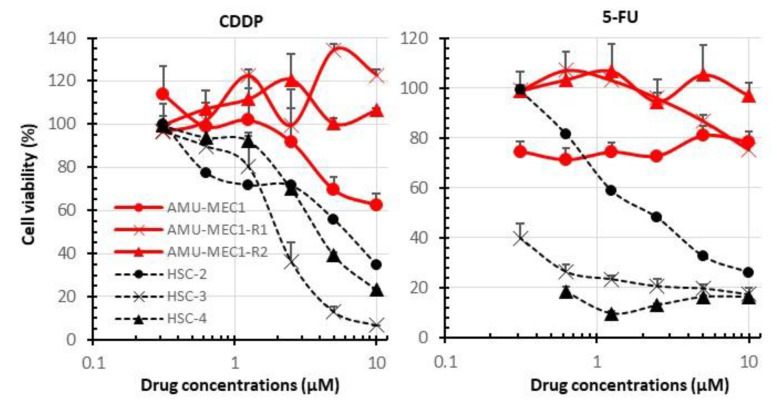
A comparison of the drug susceptibilities of MEC and OSCC cell lines. The susceptibility of each cell to CDDP (**left**) and 5-FU (**right**). The red and black dotted lines indicate MEC and OSCC cell lines, respectively. The mean and SD of triplicate measurements are shown.

**Figure 6 ijms-24-01722-f006:**
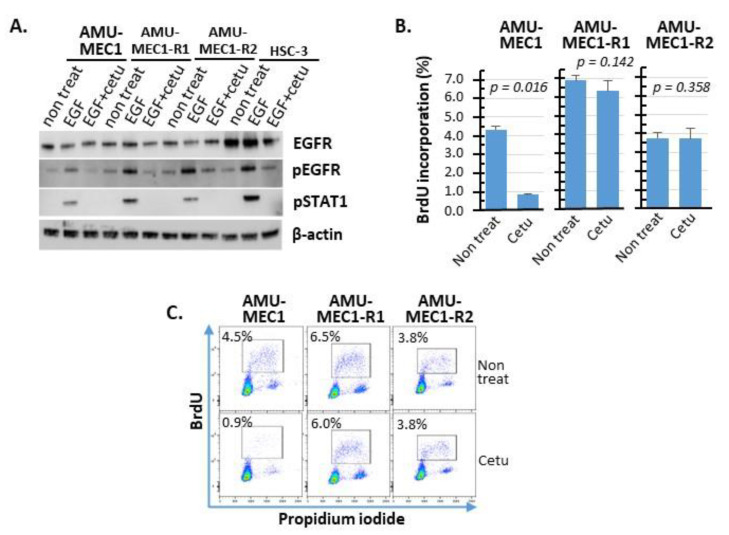
Cetuximab-mediated inhibition of EGFR signaling and EGFR-dependent cell proliferation. (**A**) Western blot analysis of EGFR expression and phosphorylation of EGFR and STAT1 in the presence of EGF or cetuximab or both in each cell. (**B**) BrdU incorporation (%) measurements performed in triplicate are shown. (**C**) Representative BrdU vs. PI cytograms for each cell line. Cetu, Cetuximab.

**Figure 7 ijms-24-01722-f007:**
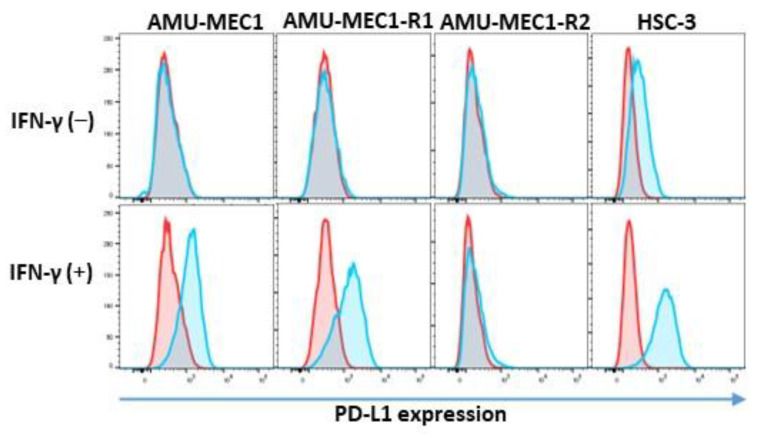
IFN-γ-induced PD-L1 upregulation in established cell lines. PD-L1 expressions in the absence of IFN-γ (upper column) and after IFN-γ stimulation (lower column) are depicted, respectively. The HSC-3 cell line served as the positive control. The red histograms indicate the isotype control, while the blue histograms indicate anti-PD-L1.

## Data Availability

The data will be available from the corresponding author upon reasonable request.

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
