# Peer review of "Establishment of Mucoepidermoid Carcinoma Cell Lines from Surgical and Recurrence Biopsy Specimens"

_ijms, 2023, doi:10.3390/ijms24021722_

Round 1

Reviewer 1 Report

Cell lines of mucoepidermoid carcinoma were obtained at the initial surgery and after recurrence. Both cell lines showed similar character in terms of morphology, bioactivity, genetic abnormalities, and poor response to chemotherapy. However, the good response to cetuximab at the initial stage was lost at the time of recurrence.

The methodology is clearly described and performed correctly. The data are also clear.

Although this is a small case report, given that mucoepidermoid carcinoma is relatively rare and we have few data about its genetic and phenotypic aspects, this paper is worth for publication. Particularly, the phenotypic change at the time of recurrence is an important finding.

Author Response

We sincerely thank you for appreciating our research.

Reviewer 2 Report

This is an interesting study that established cell lines from both surgical and recurrent biopsy specimens of the same mucoepidermoid carcinoma patient.

A few minor revisions are listed below.

Line 88: MEC1-R1 cells were not smaller than AMU-MEC1 and AMU-MEC1-R2 cells. MEC1-R1 cells appears to be the same size as AMU-MEC1 and AMU-MEC1-R2 cells in Figure 1.

Legend to Figure 6, lines 4: B. not U

Author Response

Thank you very much for the revisions.

  • We deleted sentence Line 88 “AMU-MEC1-R1 cells were smaller than AMU-MEC1 and AMU-MEC1-R2 cells.” because this sentence makes no sense.
  • We revised Figure 6 legend according to your comments.

Please refer revised manuscript.